# Mechanical Characterization and In Vitro Assay of Biocompatible Titanium Alloys

**DOI:** 10.3390/mi13030430

**Published:** 2022-03-10

**Authors:** Iustinian Baltatu, Andrei Victor Sandu, Maria Daniela Vlad, Mihaela Claudia Spataru, Petrica Vizureanu, Madalina Simona Baltatu

**Affiliations:** 1Faculty of Materials Science and Engineering, “Gheorghe Asachi” Technical University of Iasi, 41 “D. Mangeron” Street, 700050 Iasi, Romania; baltatu.iustin@yahoo.com (I.B.); sav@tuiasi.ro (A.V.S.); 2Romanian Inventors Forum, Str. Sf. P. Movila 3, 700089 Iasi, Romania; 3Biomedical Sciences Department, Faculty of Medical Bioengineering, “Grigore T. Popa” University of Medicine and Pharmacy from Iasi, 9-13 Kogalniceanu Street, 700454 Iasi, Romania; maria.vlad@umfiasi.ro; 4TRANSCEND Research Centre, Regional Institute of Oncology, Str. G-ral Henri Mathias Berthelot 2-4, 700483 Iasi, Romania; 5Public Health Departament, Faculty of Veterinary Medicine, “Ion Ionescu de la Brad” University of Life Sciences, 3 Mihail Sadoveanu Alley, 700490 Iasi, Romania; mspatarufmv@yahoo.com; 6Technical Sciences Academy of Romania, Dacia Blvd 26, 030167 Bucharest, Romania

**Keywords:** titanium alloys, in vitro assay, mechanical properties, Ti-Mo-Zr-Ta system

## Abstract

Metals that come into contact with the body can cause reactions in the body, so biomaterials must be tested to avoid side effects. Mo, Zr, and Ta are non-toxic elements; alloyed with titanium, they have very good biocompatibility properties and mechanical properties. The paper aims to study an original Ti20Mo7Zr*x*Ta system (5, 10, 15 wt %) from a mechanical and in vitro biocompatibility point of view. Alloys were examined by optical microstructure, tensile strength, fractographic analysis, and in vitro assay. The obtained results indicate very good mechanical and biological properties, recommending them for future orthopedic medical applications.

## 1. Introduction

In specific medical applications, the selection of materials and the design of the elements that incorporate them are called ‘appropriate expectations’. This term does not take into account the changes that occur in the patient’s life after implantation [1,2], but has the role of guiding the selection of technologies before implantation. It involves trying to design the most sustainable biomaterials and the best surfaces that best meet patients’ requirements [2,3,4,5].

When performing the tests necessary to assess the interactions from the biomaterial–tissue interface, the objective and subjective factors that may influence the response of the tissue, but also those of the biomaterial, are considered in order to correctly interpret the results, depending on the material factors and specifically those related to the surface [3,5,6]. The important parameters for the direct tissue–biomaterial contact [7,8] are the biocompatibility of the material chosen for the implant, the macrostructure (shape), the microstructure (surface roughness, geometry of the elevations, and depressions in the implant), the surgical implant procedure, the direct tissue–implant contact after insertion (implantation), and the time and manner of loading the implant, which may result in movements of the implant relative to the adjacent tissue and the implant support. Recently, it has become apparent that the microstructure of the implant surface significantly influences osteoblasts, as well as the amount of tissue formed at the interface. Therefore, the biocompatibility of an implant is only one of the parameters that influences the response of the tissue to the metal implant. The morphology of the surfaces and the microscopic structures also become important because the main problem that limits the application and operation of the metal implants is the lack of a viable anchoring of the implant in the tissue. On this experimental basis, studies have been performed on samples prepared according to specific standards and protocols.

The success of an implant replacement depends very much on the interface between the tissue and the synthetic material, where totally different effects can be obtained. While blood contact implants with minimal interaction are normally required for implants that remain in contact with blood, osseointegrated implants must have strong interaction in order to obtain a high adhesion force. Osteointegration can be influenced by both the structure and topography of the surface and its composition [9,10].

Orthopedic implants are in increasing demand due to the growing share of the elderly in the population of the Earth, contributing to the increase in the consumption of such implants [3]. Therefore, physicians and biomedical researchers need to be prepared to respond to the growing need for healthcare in any kind of illness or trauma and in any part of the world. Traditional biomaterials are based on polymers, ceramics, and metals; currently, the most used biomaterials are based on metals, especially titanium-based alloys [10,11,12].

Titanium alloy exhibits several quality properties, such as good low temperature ductility, good biocompatibility, corrosion resistance, mechanical load resistance, as well as the property of converting heat energy into mechanical energy. It is used in dentistry (dental implants), reconstructive surgery (cranial plaques), cardiac surgery (artificial heart), and orthopedics (fracture clamps and screws) [13,14,15,16].

The first Ti alloy to combine Ti-specific biocompatibility properties with mechanical properties at least as good as those of conventional materials is the Ti-6Al-4V alloy [17]. It is one of the most widely used titanium alloys in medicine, it has low wear resistance, a high modulus of elasticity (about 4–10 times higher than human bone), and a low shear strength. Research in the field has shown that the use of this alloy in implantology involves risks of toxic reactions due to the presence of vanadium and aluminum in the composition. Vanadium has a high cytotoxicity, and aluminum can even induce senile dementia [18,19,20].

Due to the consideration that alloying elements that are part of the alloy must be in the category of non-toxic elements, Ti-Mo alloys are increasingly studied, have adequate properties, and can be used for applications in implantology [21,22,23]. Ti-Mo alloys alloyed with various biocompatible elements have shown superior mechanical properties, such as high tensile strength and a much lower modulus of elasticity—close to that of human bone—when compared to other classical biomaterials. Studies have shown that α + β alloys have satisfactory characteristics, especially with superior mechanical properties, as well as increased resistance of α alloys to both corrosion and oxidation. On the other hand, β-type alloys—due to the stabilizing elements Mo, Ta, and Nb—have the advantages of increasing the mechanical strength and a modulus of elasticity close to that of human bone, which are important aspects regarding the long-term use of biomaterials in the medical field [24,25,26].

Many papers have performed biocompatibility studies [6,25,26,27] on Ti-Mo and C.P. Ti, indicating that these alloys are not highly toxic and are biocompatible with human tissue. Studies have highlighted that these materials did not cause any aggression, and also that no change in cell morphology or adhesion to the surface of the material was induced [28,29,30,31].

The biocompatibility of titanium and its alloys is closely related to the formation of oxide layers (TiO_2_) on the surface of the implant [32,33,34], as an interface between the implant and the biological environment. Alloy elements—such as Zr, Ta, Nb, and Sn—do not affect cell viability and have a reduced number of ions released in the body, but Al and V contribute to reduce cell viability [34,35,36]. Other elements—such as Ag, Co, Cr, and Cu—have moderate cytotoxicity behavior, but their presence in these alloys significantly reduces their toxicity [37,38].

To avoid the disadvantages of titanium alloys commonly used in orthopedic applications, non-toxic elements (Mo, Zr, and Ta) were added to the metallic matrix of pure titanium in order to enhance the mechanical and biological properties. The paper contains the study of three alloys: Ti20Mo7Zr5Ta, Ti20Mo7Zr10Ta, and Ti20Mo7Zr15Ta from a mechanical and in vitro biocompatibility point of view.

## 2. Materials and Methods

### 2.1. Obtaining of Ti-Mo-Zr-Ta

The elaboration of biomaterials has special requirements, both for the study of the alloy design parameters and for the technological parameters of the elaboration equipment. The MRF ABJ 900 Vacuum Arc Remelting equipment (MRF, Allenstown, NH, USA) was used to obtain the three alloys from the Ti-Mo-Zr-Ta system (Ti20Mo7Zr5Ta, Ti20Mo7Zr10Ta, Ti20Mo7Zr15Ta). This equipment has the advantage that very high melting temperatures can be achieved and is thus able to create alloys with uniform composition by repeated remelting. The raw material used was of high purity (Ti-99.8%, Mo-99.7%, Zr-99.2%, and Ta-99.5%) purchased from Alfa Aesar. During the melting operations, first a vacuum atmosphere of 4.5 × 10^−3^ mbar was applied, followed by purging the enclosure with inert gas (high purity Ar 5.3). This cycle was repeated four times to purify the working atmosphere in the VAR installation. For each ingot, the melting process was repeated six times, on each side, in order to refine and homogenize the chemical composition. The technological parameters values established for obtaining the experimental Ti-based alloys were the following: melting power of min. 55 kVA; melting current of min. 650 A, 60% DS, three-phase voltage; vacuum level of 4.5 × 10^−3^ mbar; inert gas flow of 5 L/min.

### 2.2. Morphological and Structural Analysis

To highlight the microstructural aspects, a microscope Zeiss Axio Imager A1 (Carl-Zeiss-Strasse, Oberkochen, Germany) was used for optical analysis for high-precision optical images. The alloys were cut to precise dimensions (10 × 10 × 5 mm), and were properly prepared on Tegramin grinding and polishing equipment (Struers Inc., Cleveland, OH, USA). DiaPro Allegro/Largo 9 µm, supplied by Struers, was used for high-performance metal grinding, which involved a stable diamond suspension containing a unique blend of high-performance diamonds and cooling lubricant. The reagent used to highlight the microstructure was: 10 mL HF, 5 mL HNO_3_, 85 mL H_2_O.

### 2.3. Tensile Testing

The INSTRON 8800 Universal Tester (100 kN) (Instron, Norwood, MA, USA) with hydraulic tanks for tensile testing was used to test Ti-Mo-Zr-Ta alloys. The samples were tested by elongation along their main axes at a constant speed until breaking. The test was performed at ambient temperature with a stress rate of 0.5 N/min, which allows the acquisition of as many essential points as possible on the characteristic curve σ-ε (stress–strain curve). Samples with a gauge length of 40 mm were used for tensile testing. The servo-hydraulic testing equipment performs tensile, compression, bending, peeling, and other mechanical tests on the materials and products according to ASTM, ISO, and other industry-specific standards [39].

In order to determine a series of mechanical characteristics that allow the assessment of the behavior of the developed alloys, samples with standardized dimensions were taken from the tested alloys respecting the national and international standards in force. The samples were subjected, at their request, by applying a progressive tensile load in the direction of the longitudinal axis, thus determining the main properties.

### 2.4. Fractographic Tests

Fractography is performed because most defective materials have cracks and surface properties, providing unique indications of how fractures occur and therefore how fragility relates to fracture mode. Each chemical analysis and mechanical test has its own sampling requirements.

After the samples of Ti-Mo-Zr-Ta alloys were tested on tensility to rupture, they were studied by fractographic analysis. A scanning electron microscopy (SEM-EDX Vega 2 LSH Tescan, Brno, Czech Republic) was used for the analysis of fracture surfaces, with the samples being prepared accordingly to the microscopic techniques used. The sensors for detecting signals in the microscope chamber can be ETD (secondary electron system), X-ray, and BSED (retro-diffused electron system). The detection system captures signals from these types of sensors, converts them into amplified electrical signals, and then sends them to the computer for processing and displaying the image on the monitor. The inspection mode of the Inspect S microscope can be switched quickly and easily with the help of a mouse cursor and the graphical operating interface of the microscope, comprising two variants: advanced vacuum (high vacuum) and preliminary vacuum (low vacuum).

### 2.5. In Vitro Cytocompatibility Tests

In order to prove the in vitro biocompatibility of the elaborated alloys, samples were taken at specific dimensions (5 × 5 mm) and evaluated by the MTT (i.e., 3-(4,5-dimethyltiazol-2-yl)-2,5-diphenyl tetrazolium bromide) colorimetric assay [38,40,41], which allows the quantification of the number of living cells, due to the cells’ ability to reduce the MTT at formazan (as dark-blue crystals) thanks to mitochondrial enzyme activity. For this, Ti-Mo-Zr-Ta alloy samples were co-incubated with HOS cells (human osteosarcoma cells; CLS, Eppelheim, Germany), by means of direct contact, for 3 and 9 days, respectively. Briefly, the cytocompatibility test was done in a 24-well plate at a cell seeded density of 1 × 105 cells/sample/well and in standard condition (i.e., incubation at 37 °C, 5% CO_2_, and a humid atmosphere of 95%). In addition, well containing only cells seeded in complete culture medium (without alloys samples) were also used as control-wells. The HOS cells were cultured in MEM culture medium (minimal Eagle’s medium; Sigma-Aldrich, Germany) supplemented with 10% fetal bovine serum (FBS), 2% L-glutamine and 1% antibiotic (penicillin-streptomycin). For the 3-day test, the culture medium was refreshed after 24 h and in the case of the 9-day test, the media was renewed for the first time after 1 day and then at established intervals of 48 h. After the specified times of co-incubation, the cells were rinsed with phosphate buffered saline solution (PBS), treated with MTT dye solution and incubated in standard condition for 3 h in order to assure the formazan crystals formation inside of viable cells. Thereafter, the dye liquid was removed, the formazan was solubilized with dimethyl sulfoxide (DMSO) under continuous agitation for 15 min, and the absorbance of each well was read by the means of a microplate reader (FilterMax F5 Multimode Microplate Reader; Molecular Devices) at a wavelength of 570 µm. The resulted viability profile (VP) for the studied alloys was expressed as percentages of the control-wells’ viability according with the formula: VP = 100 × (alloys-wells’ optical density–empty-wells’ optical density)/(control-wells’ optical density–empty-wells’ optical density). The viability study was done in triplicate and the data were expressed as mean ± standard deviation (mean ± SD; n = 3). Finally, the one-way ANOVA test was used in order to perform the statistical analysis of the viability data [42], and the Tukey’s method was applied to compare the results, with statistically significant differences being accepted at *p* < 0.05.

## 3. Results and Discussions

### 3.1. Microstructural Analysis

Titanium exists in two allotropic forms. At low temperatures, it has a closed structure of hexagonal crystal (cph), which is known as α, as above 883 °C it has a cube structure with centered faces called β. The transition temperature of pure titanium from α to β increases or decreases according to the properties of alloying elements. Alloying elements that tend to stabilize the α phase (such as Al, O, N, etc.) are called α stabilizers. Adding these elements will increase the β temperature, while the elements that stabilize the β phase are called stabilizers (V, Mo, Nb, Fe, Cr, etc.), adding these elements will lower the β temperature. Some elements that have no significant effect on the stability of any phase, but that form a solid solution with titanium, are called neutral elements (Zr and Sn).

The phases α and β are also the basis for the generally accepted classification of titanium alloys. An alloy with only α stabilizer and completely composed of α phase is called α alloy. An alloy with a variety of β stabilizers, in which the β phase can be retained by rapid cooling, is called a metastable β alloy. Over time, these alloys decompose into α + β. Most biomedical titanium alloys belong to the α+β or metastable β category.

In all conventional titanium alloys, evolution from α to β plays an important role, with a major contribution on microstructure and mechanical properties. Alloys with an acicular or lamellar structure are usually known as β structure.

Figure 1 shows the resulting microstructure of the alloys in the Ti-Mo-Zr-Ta system. In Figure 1a can be observed large grains boundary of beta and alpha colonies inside them. Figure 1b,c illustrates a predominantly beta microstructures with a lamellar structure due the increased concentration of Ta (β element).

The high percentage of beta-stabilizing elements in this group of titanium alloys results in a microstructure that is metastable beta.

### 3.2. Tensile Strength Test

The appreciation of the utility of a metallic material in the technical field is made based on the knowledge of its properties. Along with the chemical and physical properties, it is necessary to know the mechanical and technological properties, which provide indications on its behavior in operation as well as on the processing possibilities. The study of materials is important because these properties depend on the internal structure of the material. The main purpose of these determinations is to observe the dependence of the mechanical and technological properties on the internal structure of the material, the structure of which was initially analyzed with a metallographic microscope.

The Ti-Mo-Zr-Ta samples were tested for tensile strength test by applying a progressive tensile load in the longitudinal axis. Figure 2 illustrates characteristic stress/strain curves and in Table 1 are presented the results data obtained of investigated alloys. These results allow an assessment to be made of the behavior of the material.

Static testing of the investigated materials in Ti-Mo-Zr-Ta consisted of an external stress, a ‘force’ that was applied to the specimens slowly, progressively, from zero to the final value, generally to rupture.

The tensile strength values for Ti-Mo-Zr-Ta alloys ranged from 712.28 MPa (Ti20Mo7Zr15Ta) to 1365.27 MPa (Ti20Mo7Zr5Ta). Thus, an average tensile strength of 1043.09 MPa was obtained, the values are inversely proportional to the increase in the percentage of tantalum. Comparing with the literature (316L: 500–1350 MPa, Co-Cr-Mo: 900–1000 MPa), the Ti-Mo-Zr-Ta alloys have higher values, which proves that the developed alloys have very good mechanical properties good.

The modulus of longitudinal elasticity (E = tgα) depends to a small extent on the microscopic structure of the alloy and on the processing technology, largely given by the type of crystal lattice. The modulus of elasticity of the alloys obtained by the tensile test was between 36.39 GPa (Ti20Mo7Zr15Ta) and 46.50 GPa (Ti20Mo7Zr5Ta). Due to the increase in the percentage of tantalum, there is a tendency for the modulus of elasticity to decrease. Recently, β-titanium alloys have become one of the most current topics in the field of biomedical titanium alloys for lower modulus of elasticity, compared to α + β titanium alloys. Ti-Mo-Zr-Ta alloys have a lower modulus of elasticity of up to 47 GPa, which is close to that of bone (below 20 GPa), while the modulus of elasticity of stainless steel can reach 193 GPa or Co-based alloys around 200 GPa.

The mechanical properties decide the type of material that will be selected for a particular application. Some of the most important properties are hardness, tensile strength, modulus, and elongation. A material’s response to repetitive cyclic loads or strains is determined by the material’s fatigue strength, and this property determines the long-term success of the implant under cyclic loading. If the implant breaks due to insufficient strength or mechanical property mismatch between the bone and the implant, this is called biomechanical incompatibility. The bone material should have a modulus equivalent to that of bone (approximately 20 GPa). Existing implant material is stiffer than bone, preventing necessary stress transmission to adjacent bones, leading to bone resorption around the implant and thus weakening of the implant. This biomechanical incompatibility that causes bone cell death is called the ‘stress protection effect’. Therefore, materials with a good combination of high strength and low modulus closer to bone should be used for implantation to avoid weakening of the implant and to prolong the service life and avoid revision surgery.

These properties recommend Ti-Mo-Zr-Ta alloys as ideals for medical applications, especially for orthopedic surgery.

### 3.3. Fractographic Analysis

In Figure 3a and Figure 4b (magnification power 100×), a macroscopic image of the breaking area of the Ti20Mo7Zr5Ta and Ti207Zr10Ta alloys is presented. Mixed breaking areas (ductile and brittle) and stretching/neck breaking areas are observed. Figure 3b and Figure 4b shows a detail on a ductile fracture area. These types of breakage are very common in titanium alloys [43].

Figure 5a shows a macroscopic image (magnification power 100×) of the Ti20Mo7Zr15Ta alloy breaking zone were mixed breaking zones (ductile and brittle) are observed. Figure 5b presents a detail on a predominantly brittle fracture zone.

The Ti-Mo-Zr-Ta alloys analyzed are characterized by the mixed breaking mode (ductile and brittle), some of them being predominant in ductile breaking (Ti20Mo7Zr5Ta) and others in brittle breaking (Ti20Mo7Zr15Ta). The rupture aspect shows a tendency to form pre-rupture material detachments, crack-like, bordered by ductile fracture zones (Ti20Mo7Zr5Ta).

### 3.4. In Vitro Cytocompatibility of Ti-Mo-Zr-Ta Alloys

The viability profile obtained for the studied Ti-Mo-Zr-Ta alloys after 3 and 9 days are shown in Figure 6, being expressed as a percentage of the control-wells’ cell viability. HOS osteoblast-like cells (cell line derived from human osteosarcoma, phenotypically close to osteoblasts) were used in this study as cellular models for osteogenic cells, well characterized and validated to test the biocompatibility of many biomaterials [44,45,46,47].

The viability data recorded after 3 days seem to suggest that by increasing the amount of Ta used for alloying, the cytocompatibility was improved significantly, being higher in the case of the Ti20Mo7Zr15Ta alloy compared to the Ti20Mo7Zr5Ta alloy (*p* < 0.05). In addition, the level of viability obtained after 9 days of co-incubation was not significantly different (*p* > 0.05) in the case of all the studied alloys, and also in comparing the viability data after 3 and 9 days.

However, despite the slightly lower viability level both after 3 and 9 days of co-incubation (i.e., ≥70%) compared with the control-wells (as negative control) we can appreciate (according to ISO 10993-5, [48]) that the studied Ti-Mo-Zr-Ta alloys do not have a cytotoxic effect, and this is so because during the study the cells were directly exposed to the alloy samples (up to 9 days) and consequently to an 100% extract due to the fact that the culture media was renewed every 48 h. Furthermore, the cytocompatibility data obtained in this study do not in any way contraindicate [48] the optimal in vivo behavior of the experimental Ti-Mo-Zr-Ta alloys and the potential use in biomedical applications. In this sense, it should be pointed out that the interface between a biomaterial and a body tissue is quite complex, which can lead to the long-term success of the implant, under a complete understanding and evaluation of the events that take place at these interfaces, i.e., the chemical and ionic distribution on the surface that may deliberately induce effects necessary to mediate the in vivo cellular response and the implants’ further osteointegration. In this sense, it should be pointed out that the in vivo interaction between biomaterials and bone cells/tissue is always more complex than in laboratory experiments (i.e., static conditions with the risk of ions’ concentration increase in the microenvironment and subsequent inhibitory effect on cell growth), which can lead to the long-term success of the implant in animal models and/or real biomedical applications under hydrodynamic conditions (i.e., body fluid flow and mass transfer at bone–implant interface); i.e., the chemical and ionic distribution on the surface may induce the suitable effects to mediate the in vivo cellular response and the implants’ further osteointegration.

## 4. Conclusions

This paper presents aspects related to structural and mechanical properties, as well as biocompatibility of three alloys: Ti20Mo7Zr5Ta, Ti20Mo7Zr10Ta, and Ti20Mo7Zr15Ta.

The structure of the investigated alloys was specific to β-type alloys, due to the presence of Mo and Ta alloying elements in the composition of titanium alloys. The Ti20Mo7Zr5Ta alloy showed a structure with large β-type grains while the other alloys (Ti20Mo7Zr10Ta and Ti20Mo7Zr15Ta) showed a dendritic structure.

The mechanical properties provided by the tensile testing showed an average tensile strength of 1043.09 MPa and a low modulus of elasticity, which indicates that the developed alloys have good mechanical properties compared to other biomaterials on the market which suggests their usefulness in future orthopedic applications.

Fractographic analysis provided aspects of alloy breaking during the tensile test, these being characterized by the mixed breaking mode (ductile and brittle), with some of them predominantly demonstrating ductile breaking (Ti20Mo7Zr5Ta).

In vitro tests performed on alloys investigated using the MTT method indicated a viability level ≥70% both after 3 and 9 days of co-incubation compared with the control culture. The investigated alloys are non-toxic and can be used in future in vitro studies and subsequent biomedical applications.

## Figures and Tables

**Figure 1 micromachines-13-00430-f001:**
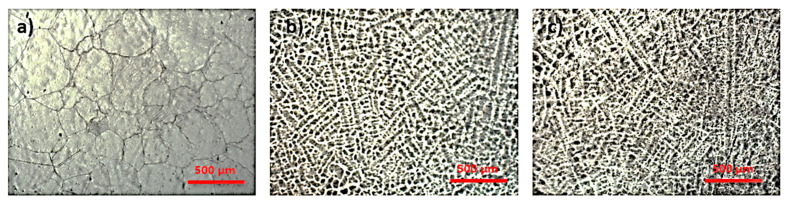
Optical microscopy images: (**a**) Ti20Mo7Zr5Ta, (**b**) Ti20Mo7Zr10Ta, (**c**) Ti20Mo7Zr15Ta.

**Figure 2 micromachines-13-00430-f002:**
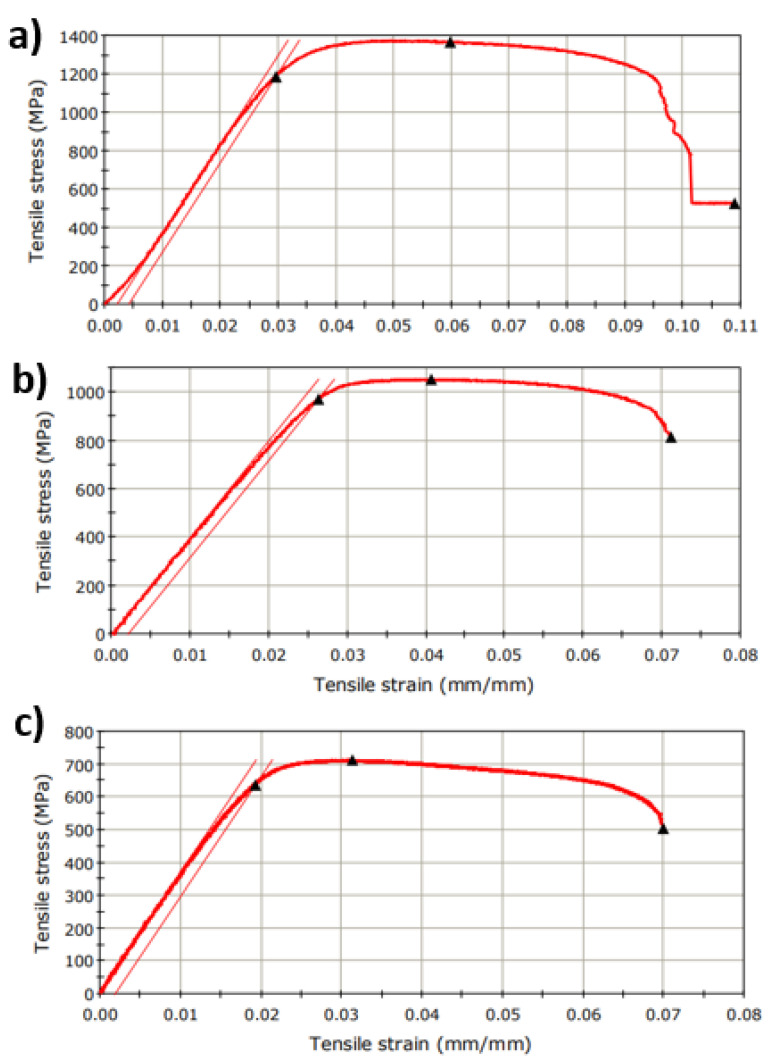
Stress–strain curves obtained on investigated alloys: (**a**) Ti20Mo7Zr5Ta, (**b**) Ti20Mo7Zr10Ta, (**c**) Ti20Mo7Zr15Ta.

**Figure 3 micromachines-13-00430-f003:**
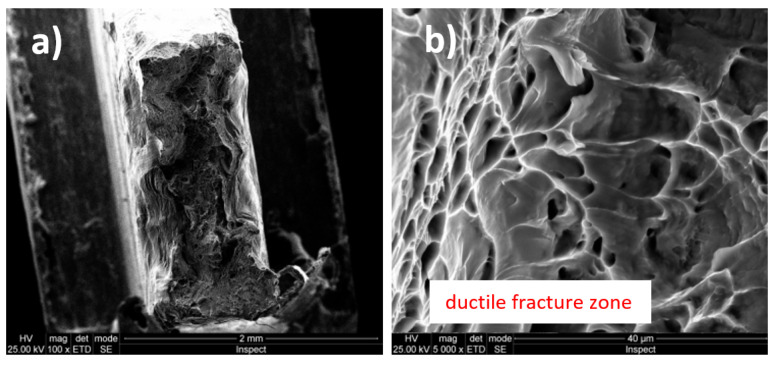
Fracture surfaces (fractography) images of Ti20Mo7Zr5Ta alloy: (**a**) 100×; (**b**) 5000×.

**Figure 4 micromachines-13-00430-f004:**
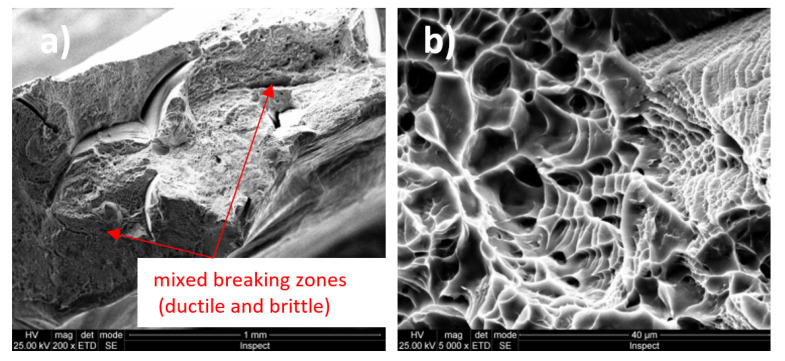
Fracture surfaces (fractography) images of Ti20Mo7Zr10Ta alloy: (**a**) 200×; (**b**) 5000×.

**Figure 5 micromachines-13-00430-f005:**
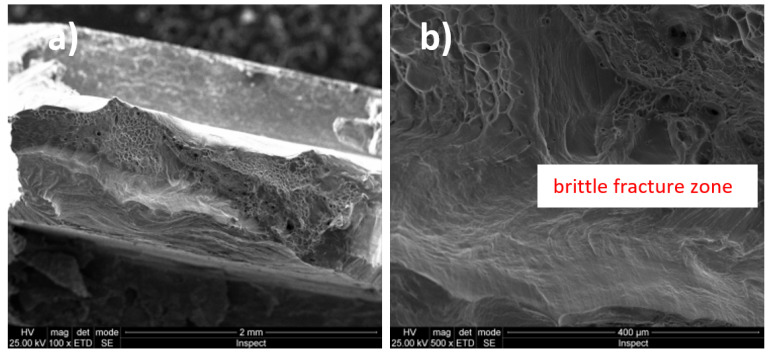
Fracture surfaces (fractography) images of Ti20Mo7Zr15Ta alloy: (**a**) 100×; (**b**) 500×.

**Figure 6 micromachines-13-00430-f006:**
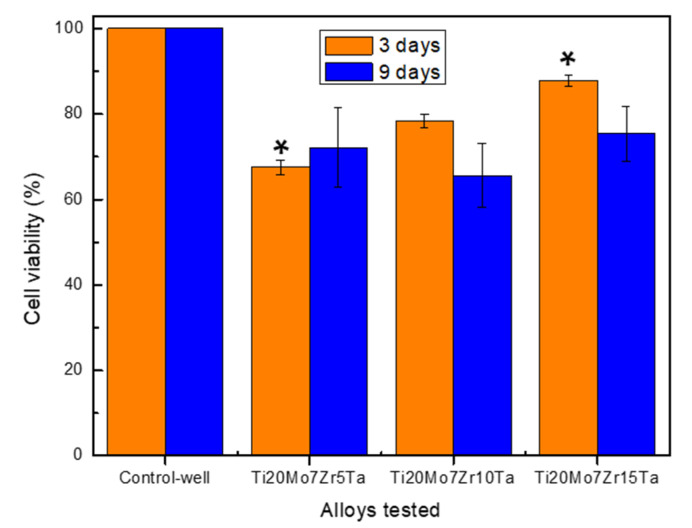
Cell viability profile (%; mean ± SD; n = 3) of HOS cells evaluated by MTT-assay: effect of Ti-Mo-Zr-Ta alloy samples on HOS cells’ viability after 3 and 9 days of culture (i.e., 3D and 9D). No significant differences on viability profile (*p* > 0.05) between alloys samples after 9D. (∗) Significant differences (*p* < 0.05) between them at 3D (see details in the text).

**Table 1 micromachines-13-00430-t001:** Summary of tensile properties on investigated alloys.

Characteristics	Ti20Mo7Zr5Ta	Ti20Mo7Zr10Ta	Ti20Mo7Zr15Ta
Extension at Tensile Strength (mm)	0.91	0.60	0.51
Load at Tensile Strength (N)	4616.39	3617.76	2511.38
Tensile Strain at Tensile Strength (mm/mm)	0.06	0.04	0.03
Tensile Stress at Tensile Strength (MPa)	1365.27	1051.74	712.29
Energy at Break (Standard) (J)	5.86	2.97	2.35
Load at Break (Standard) (N)	1788.90	2802.12	1779.06
Extension at Break (Standard) (mm)	1.65	1.04	1.15
Tensile Strain at Break (Standard) (mm/mm)	0.11	0.07	0.07
Tensile Stress at Break (Standard) (MPa)	529.06	814.62	504.58
Modulus (Automatic Young’s) (MPa)	46,508.25	40,123.67	36,390.64
Load at Yield (Offset 0.2%) (N)	4011.49	3336.26	2242.77
Tensile Strain at Yield (Offset 0.2%) (mm/mm)	0.03	0.0003	0.02
Tensile Stress at Yield (Offset 0.2%) (MPa)	1186.37	969.90	636.10
Area (mm^2)	3.38	3.44	3.53
Geometry	Rectangular		
Length (mm)	15.20	14.68	16.40
Thickness (mm)	1.17	1.17	1.22
Width (mm)	2.89	2.94	2.89
Elongation (%)	10.85	7.08	7.01

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
