# Peer review of "Mechanical Characterization and In Vitro Assay of Biocompatible Titanium Alloys"

_micromachines, 2022, doi:10.3390/mi13030430_

Round 1
Reviewer 1 Report
The work concerns the research of three titanium alloys. The microscopic examinations, static tensile tests, fractographic investigations and biocompatibility tests were carried out. The description of the research is very laconic and in many places incorrect. The work should be generally rejected, but in order to give the Authors a chance to improve it - refer it for major revision with an indication that it requires a very deep reconstruction.
Detailed comments:
Line 106: Please indicate the location of the manufacturer of all devices in accordance with the journal guidelines
Line 113: Please indicate the remelting temperature
Line 119: Please indicate the type of polishing paste used
Line 125: Please indicate the strain rate
Figure 1: The description of the microstructure is very brief. The microstructure of the Ti20Mo7Zr5Ta alloy is clearly different from Ti20Mo7Zr10Ta and Ti20Mo7Zr15Ta. The last two are clearly dendritic in nature. This requires a comment. At the same time, the applied magnification does not allow the assessment of the microstructure. It is required to present the microstructure at higher magnification.
Figure 2: I propose to combine the figures in one graph. Currently, it is difficult to compare them. What is the result of the plateau that appears before the final break of the sample?
Table 1. Please use the correct strength parameters. Relative values [in%] should be given for elongation. How to understand negative values in a table? Do not completely believe in the parameters set by the software.
Line 227-231: It is too general to say that the tested alloys have better properties. In particular with regard to the reported values for steel and cobalt alloy, which are indicated in a very wide range. Taking into account the dendritic microstructure of the alloy, the obtained values are unlikely.
Line 233-241: Titanium has a Young's modulus of 105-120 GPa. It is not possible for its alloys to have so different value of this parameter.
Figure 3-5: Please indicate areas of brittle fracture in the figures. In my opinion, the fracture is ductile. It is also indicated by the nature of the curve obtained in the static tensile test.
Line 324: The brittle nature of the fracture (if any) practically excludes the alloy from orthopedic applications.
Author Response
Thank you very much for suggestions and recommendations. We have done the correction of the manuscript accordingly.

Reviewer 2 Report
This manuscript describes and characterizes a titanium alloy system that could be potentially used in orthopedic medical applications. The new alloy system shows several good mechanical and biological properties. Overall, the reported alloy system could be interesting to the community, however, there are a couple weaknesses of the manuscript. They should be addressed before being considered for publication.
Major:
1) Expert editorial review is recommended to correct many grammatical errors and wording issues.
2) A very important experiment in this manuscript is the cell viability test. However, it seems like the cell viability drops remarkably in 9 days to a level around 70%. The author should offer a clearer explanation about why this is still an acceptable biocompatibility in the context for potential orthopedic applications, and why a 9-day period is long enough for such tests.
The author may also investigate the cell attachment/adhesion to the alloy surface via fluorescence/stain methods.
3) Figure 3-5 are all fractographic images of three alloy compositions. Is it necessary to have 4 different imaging magnifications for each composition? The author may want to combine figure 3-5 as a comprehensive fractographic figure.
Minor:
1) Page 1, in the abstract the author mentions XRD, but it seems like this technique is not used in the manuscript.
2) Page 1, line 39, The author may want to add references for the parameters that are important to tissue- biomaterial contact.
3) Page 3, line 127. Please add references for the standards mentioned for tensile testing. What is the elongation rate used and what is the standard sample size?
4) Page 4, line 161, please add vendor information for the cell culture medium; line 175, add reference for the ANOVA test. Is the cytocompatibility test measured in triplicate?
5) How the differed micro-structures observed in figure 1 is related to temperature resistance described in the main text?
6) Please explain the meanings of the linear fittings in Figure 2.
7) In Figure 6, please show the fill size of the error bar and indicate how the error bars are calculated. There is no need to have ticks on the top x-axis and right y-axis.
Author Response

(The authors gave the same response as above.)

Reviewer 3 Report
How do authors confirm the phase transformation in alloy using SEM analysis? Need significant evidence for Section 3.1.
Figure caption for Fig 3 and Fig 4 are similar in citation, authors can rewrite the same.
Mark the features in fractured surface of the test samples. Indicate the type of fracture in the manuscript with literatures.
The invitro test analysis needs a detailed discussion. What type of cell and its morphological study is expected for the invitro analysis?
Author reported about MTT assay in this paper. What is the rate of growth on cell for increase in time? Authors can refer this paper for idea - https://doi.org/10.1007/s40735-020-0326-5.
Author Response

(The authors gave the same response as above.)

Round 2
Reviewer 1 Report
The manuscript has only been slightly revised. The main complaints still remain valid.
The way of presenting the strength parameters indicates that the Authors do not use them correctly. The parameters were not presented in accordance with the principles of tensile tests. Please consult an expert in this regard. A different value of the sample geometry is indicated in the Table 1 and another in the research methodology. Was the elongation determined at the length of 15 mm, 40 mm or was the sample proportional? How was the energy at break parameter [J] obtained?
Where do the Authors notice the lamellar microstructure on Figure 1? It is still unclear why there are such large differences in the microstructure of higher tantalum alloys compared to the Ti20Mo7Zr5Ta alloy. If tantalum is a beta phase stabilizer, its content should be highest for an alloy containing 15% Ta. Meanwhile, the microstructures in Figures 1b and 1c are similar.
The presence of two separate fracture zones for the Ti20Mo7Zr15Ta alloy indicates the material heterogeneity. This is not a typical brittle-plastic fracture.
I cannot accept the work as it is.
Reviewer 2 Report
Overall the author has addressed my comments properly, though the quality of the figures should be further improved. The texts in the figures are not clear to read and the numbering in figure 3,4,5 are not right.
For the cell viability test, it might be better to include a photo showing the experimental setup in figure 6.